# Automatic Image Registration Provides Superior Accuracy Compared with Surface Matching in Cranial Navigation

**DOI:** 10.3390/s24227341

**Published:** 2024-11-18

**Authors:** Henrik Frisk, Margret Jensdottir, Luisa Coronado, Markus Conrad, Susanne Hager, Lisa Arvidsson, Jiri Bartek, Gustav Burström, Victor Gabriel El-Hajj, Erik Edström, Adrian Elmi-Terander, Oscar Persson

**Affiliations:** 1Department of Clinical Neuroscience, Karolinska Institutet, SE 17177 Stockholm, Sweden; henrik.frisk@ki.se (H.F.); margret.jensdottir@ki.se (M.J.); lisa.arvidsson@ki.se (L.A.); jiri.bartek@ki.se (J.B.J.); gustav.burstrom@ki.se (G.B.); victor.gabriel.elhajj@stud.ki.se (V.G.E.-H.); erik.edstrom.1@ki.se (E.E.); oscar.persson.1@ki.se (O.P.); 2Department of Neurosurgery, Karolinska University Hospital, SE 17176 Stockholm, Sweden; 3Clinical Affairs, Brainlab AG, 81829 Munich, Germany; luisa.coronado@brainlab.com (L.C.); markus.conrad@brainlab.com (M.C.); susanne.hager@brainlab.com (S.H.); 4Capio Spine Center Stockholm, Löwenströmska Hospital, SE 19489 Upplands-Väsby, Sweden; 5Department of Surgical Sciences, Uppsala University, SE 75236 Uppsala, Sweden

**Keywords:** patient tracking, reference frame, surface matching, CBCT, neurosurgery, surgical navigation, automatic image registration, accuracy

## Abstract

Objective: The precision of neuronavigation systems relies on the correct registration of the patient’s position in space and aligning it with radiological 3D imaging data. Registration is usually performed by the acquisition of anatomical landmarks or surface matching based on facial features. Another possibility is automatic image registration using intraoperative imaging. This could provide better accuracy, especially in rotated or prone positions where the other methods may be difficult to perform. The aim of this study was to validate automatic image registration (AIR) using intraoperative cone-beam computed tomography (CBCT) for cranial neurosurgical procedures and compare the registration accuracy to the traditional surface matching (SM) registration method based on preoperative MRI. The preservation of navigation accuracy throughout the surgery was also investigated. Methods: Adult patients undergoing intracranial tumor surgery were enrolled after consent. A standard SM registration was performed, and reference points were acquired. An AIR was then performed, and the same reference points were acquired again. Accuracy was calculated based on the referenced and acquired coordinates of the points for each registration method. The reference points were acquired before and after draping and at the end of the procedure to assess the persistency of accuracy. Results: In total, 22 patients were included. The mean accuracy was 6.6 ± 3.1 mm for SM registration and 1.0 ± 0.3 mm for AIR. The AIR was superior to the SM registration (*p* < 0.0001), with a mean improvement in accuracy of 5.58 mm (3.71–7.44 mm 99% CI). The mean accuracy for the AIR registration pre-drape was 1.0 ± 0.3 mm. The corresponding accuracies post-drape and post-resection were 2.9 ± 4.6 mm and 4.1 ± 4.9 mm, respectively. Although a loss of accuracy was identified between the preoperative and end-of-procedure measurements, there was no statistically significant decline during surgery. Conclusions: AIR for cranial neuronavigation consistently delivered greater accuracy than SM and should be considered the new gold standard for patient registration in cranial neuronavigation. If intraoperative imaging is a limited resource, AIR should be prioritized in rotated or prone position procedures, where the benefits are the greatest.

## 1. Introduction

During the last decades, neuronavigation has become the standard for intracranial tumor surgery. Conventional neuronavigation systems rely on the tracking of a dynamic reference frame (DRF) with a fixed relation to the patient’s head. Tracking is commonly performed using infrared cameras recognizing reflective spheres on the DRF and navigated instruments. To allow accurate navigation, the position of the patient’s anatomy must be correctly aligned with the DRF in a process called registration. Historically, registration has been performed by the acquisition of anatomical landmarks or by using adhesive skin fiducials, implanted cranial fiducials, or surface matching (SM) of the face [1,2,3,4]. The accuracy of these methods is roughly equal, with a slight disadvantage for the use of anatomical landmarks alone and a slight advantage for the use of implanted cranial fiducials [4,5]. However, the latter method has largely been abandoned due to the requirement for preprocedural implantation of fiducials and additional imaging. Most commercially available systems use surface matching for registration, either alone or in combination with the acquisition of anatomical landmarks. However, recent studies have shown that intraoperative CT scans for automatic registration result in higher accuracy than surface matching [2,3,6,7,8].

The accuracies of different neuronavigation systems have been investigated in several studies [5,9,10,11,12,13,14,15,16] and are reported to be in the range of 0.5–5 mm [17,18]. However, these studies mostly use skull phantoms or cadaveric specimens in a laboratory environment, a setting not necessarily comparable to a real-life surgical setting. Phantoms are rigid and not prone to skin indentation by pointer tools when acquiring surface landmarks or skin deformation by gravitational forces in rotated or prone positions [19]. SM in a prone position is often challenging since the head clamp may obscure the line of sight of the tracking cameras during registration. Thus, navigational accuracy is often lower in a real-life surgical setting compared with experimental setups using phantoms or cadavers [17].

The use of automatic image registration (AIR) alleviates the problem of registration in troublesome positions and theoretically also produces a more homogenous registration accuracy within the navigated volume, independent of distance to fiducials or surfaces used for registration. The newly released Registration Matrix CT Cranial and corresponding software for AIR v.2.5 (Universal AIR, Brainlab AG, Munich, Germany) is an intraoperative 3D imaging-based solution independent of the scanning device. A registration matrix (Universal AIR) containing radio-opaque markers and infrared reflective spheres is placed in the CT imaging field. Using the known geometry of the registration matrix, a linear transformation is calculated from the image coordinate system to the registration marker coordinate system and from the registration marker coordinate system to the DRF coordinate system. Once the registration is completed, the registration matrix can be removed from the surgical field, and patient tracking relies on the DRF of the navigation system. The aim of this study was to evaluate AIR with the Universal AIR registration device using CBCT for cranial neurosurgical procedures and compare the registration accuracy with the traditional SM registration method. As a secondary outcome, the preservation of navigation accuracy throughout the surgery was investigated.

## 2. Materials and Methods

### 2.1. Study Design and Setting

This is a prospective, interventional, non-randomized study with consecutive patient enrollment to compare two different registration methods for surgical navigation in cranial neurosurgery. All adult patients (≥18 years) who were scheduled for intraparenchymal tumor surgery were eligible for inclusion in the study. Prior to inclusion, all patients signed a written consent form. The study hospital is a publicly funded tertiary care center that serves a region of approximately 2.3 million inhabitants and is the only neurosurgical center in the region. The study was approved by the National Ethical Review Board (DNR: 2020:05857).

### 2.2. Intraoperative Procedure

The study was conducted in a hybrid OR equipped with a Maquet surgical table (Alphamaquet 1150, Maquet AG, Rheinfelden, Switzerland) connected to a motorized ceiling-mounted C-arm flat detector system, enabling high-resolution cone-beam computed tomography (CBCT, AlluraClarity FD20, Philips Healthcare, Best, The Netherlands). All patients were under general anesthesia with the head fixed in a radiolucent carbon fiber head clamp (Doro, Black Forest Medical Group, Cape Coral, FL, USA) with radiolucent skull pins. To avoid metal artifacts in the intraoperative CBCT images, a radiolucent PEEK (Polyether ether ketone) DRF (Drape-Link, Brainlab AG, Munich, Germany) was attached to the Mayfield clamp.

### 2.3. Acquisition of Screw Head Points

#### 2.3.1. Surface Matching

After patient positioning, an SM registration based on preoperative MRI images was performed with the Curve Neuronavigation system (Brainlab AG, Munich, Germany, Curve 1.0) using the SoftTouch and/or Z-touch tools. Accuracy was verified on anatomical landmarks before the registration was accepted. After sterilizing, but before draping the surgical site, four small stab incisions were performed in line with the planned surgical incision, and four micro screws were inserted into the skull bone. The screws are ideal landmarks for accurate measurements of the registration methods because of their fixed position and easily recognized heads with a well-defined center, which is the cross hair of the slit. The slit cross hair in the centers of the inserted screw heads was identified and captured with a navigated pointer, and the acquired screw head points were saved in the software v. 3.5 (SM pre-drape).

#### 2.3.2. Automatic Image Registration

After the SM procedure, an AIR was performed. The Universal AIR device was positioned close to the surgical field using a radiolucent carbon fiber holding arm. A CBCT scan was performed, and the images were transferred to the Brainlab Navigation System for AIR (Figure 1). The registration accuracy was manually verified using anatomical landmarks. Using the image fusion software of the navigation workflow, the CBCT images were fused to the preoperative MRI images. The slit cross hairs of the screw heads were acquired, labeled, and stored (AIR pre-drape).

After sterile draping, including removal of the initial DRF and replacement with a sterile DRF clamped to the same drape-link fixation device, the incision was extended, and the skin flap was opened. Screw head positions were then acquired a third time (AIR post-drape). The surgery was subsequently performed according to routine procedure using the AIR for intraoperative navigation. Screws localized within the bone flap were removed. After completed surgery and before closing the wound, the screw head positions of the remaining screws were acquired a fourth time (AIR post-resection). The screws were then removed, and the skin was closed.

### 2.4. Registration Accuracy Calculations

#### 2.4.1. Accuracy Analysis of Image Fusion Algorithm (For SM Registration Accuracy Assessment Only)

An estimate was performed of the possible error contribution of image fusion of the preoperative MRI and the intraoperative CBCT scan, which was necessary for the SM TRE calculation. Foramina of the skull base were used, as these structures are recognizable in both CBCT and MRI. For each of the 11 patients, 3–7 foramina were defined in each image set. The error contributed by image fusion could then be calculated as the residual differences between coordinates of the defined foramina after image fusion. Using this method, the average error contribution by image fusion was estimated at 1.7 ± 1.0 mm (range 1.2–2.5 mm).

#### 2.4.2. Surface Matching Versus Automatic Image Registration

After surgery, the slit cross hairs of the screw heads were identified on the intraoperatively acquired CBCT scans, labeled, and stored in the software as screw head reference points (Figure 2). The accuracies of the two registration methods were measured as the target registration error (TRE), which is the discrepancy between the acquired points and their corresponding screw head reference points. The calculation required that the acquired points and reference points be defined in the same coordinate system. A transformation was performed using the fusion transformation matrix to align the CBCT scan to the preoperative MRI coordinate system. The mean TRE was calculated for each registration method to allow comparison.

### 2.5. Analysis of Patient Positioning and Preservation of Accuracy

The effect of patient positioning was evaluated. To allow this, the head positions were dichotomized into supine with the head rotated less than 45° or any position with the head rotated more than 45°. Preservation of accuracy of the registration throughout the different phases of the surgery was investigated by comparing AIR pre-drape, AIR post-drape, and AIR post-resection. Surgeries were performed by four different surgeons. The pairwise distance between screws is independent of the accuracy of the registration and was used as an estimate of how accurately the landmarks can be acquired by the surgeon [5]. No difference in registration accuracy could be observed between surgeons, but the sample size was too small to perform a statistical analysis.

### 2.6. Post Hoc Analysis: Effect of Skin Deformation on Surface Matching Accuracy

Since the TRE of the SM registration was consistently larger than in AIR, a post hoc analysis was performed to estimate the impact of skin deformation on registration accuracy.

First, a quantitative analysis of the differences between the 3D skin surface of the preoperative MRI and the intraoperative CBCT was performed. The skin surfaces of the two scans were aligned using Brainlab’s image fusion algorithm, and the data were subsequently loaded into the open-source software CloudCompare v.2.13.2. A color-coded heatmap was generated of the skin surface of the patient in the preoperative MRI relative to the one in the intraoperative CBCT.

Second, a re-calculation of the SM registration was performed by fitting the acquired surface registration points onto the skin surface of the intraoperative CBCT-based 3D model to evaluate and calculate the effect of skin shift on registration accuracy.

### 2.7. Statistical Analysis

#### 2.7.1. Superiority Analysis of TRE for Surface Matching and Automatic Image Registration

A superiority test was performed using a one-sided paired *t*-test comparing the TREs of SM to the TREs of AIR. Normal distribution was confirmed using the Shapiro–Wilk test. The threshold for statistical significance level was set to 0.01.

#### 2.7.2. Statistical Analyses of Patient Positioning, the Different Phases of the Surgery, and Preservation of Accuracy

To assess the impact of patient head positioning on registration accuracy, a hypothesis test was conducted comparing groups with head rotation less than 45° and those rotated more than 45°. Levene’s test was used to evaluate the equality of variances across these groups. Additionally, the Welch’s *t*-test, which does not assume equal variances between groups, was used to compare means.

To assess the statistical significance of differences in the mean TRE between the various phases of the surgery (AIR pre-drape, AIR post-drape, and AIR post-resection), the Friedman test, a non-parametric statistical method, was conducted. Subsequently, whenever significant differences were detected, the Dunn post hoc test with Bonferroni correction was employed to compare pairwise differences in accuracy between the different phases of the surgery.

#### 2.7.3. Statistical Analysis of Skin Deformation Data

To determine the effect skin deformation on the TRE accuracy, values from the quantitative analysis in Section 2.6 were used. The one-sided Wilcoxon Signed-Rank test was used to test for superiority.

## 3. Results

### 3.1. Accuracy Comparison of SM to AIR

Registration accuracy data from 22 patients undergoing cranial neurosurgery was included. The mean target registration error (TRE) for SM pre-drape was 6.6 ± 3.1 mm and 1.0 ± 0.3 mm for AIR pre-drape (Figure 3). The AIR was significantly superior to the SM registration (*p* < 0.0001), with a mean improvement in TRE of 5.58 mm (3.71–7.44 mm 99% CI) (Figure 4).

The maximum and minimum TRE for SM were 12.36 mm and 0.79 mm (99% CI of the mean, 4.71 mm–8.51 mm). The maximum and minimum TRE for AIR were 1.65 mm and 0.60 mm, (99% CI of the mean, 0.86 mm–1.21 mm) (Figure 5).

### 3.2. Effects of Patient Positioning and Preservation of Accuracy

We found very small variations in pairwise distances between screws, indicating a high accuracy in acquiring the defined targets at the centers of the screw heads.

When SM TREs were analyzed in relation to the patient’s head position, there was no significant difference in registration accuracy between head positions with rotations less than 45° (n = 9) and those with rotations greater than 45° (n = 13; *p* = 0.254). Although a tendency toward decreased registration accuracy with greater head rotation was visually noted (Figure 6), the statistical test did not confirm this observation as statistically significant (*p* > 0.05). Nonetheless, all but one patient with an SM TRE > 6 mm was in the prone position or had a head rotation beyond 45° (Figure 6). No patient with a prone position had a mean SM TRE less than 5 mm.

The mean TRE for AIR pre-drape was 1.0 ± 0.3 mm. The corresponding TREs post-drape and post-resection were 2.9 ± 4.6 mm and 4.1 ± 4.9 mm, respectively. The statistical analysis revealed a significant difference in TRE between the groups (*p* = 0.0001). Although the boxplot suggested a decline in TRE between AIR pre-drape and AIR post-drape, subsequent post hoc analysis indicated no significant difference in TRE between AIR pre-drape and AIR post-drape (*p* = 0.06), as well as no significant difference in TRE between AIR post-drape and AIR post-resection (*p* = 0.13). However, a statistically significant difference in TRE was observed between AIR pre-drape and AIR post-resection (*p* = 0.00005).

### 3.3. Effect of Skin Deformation on Surface Matching Accuracy

Fusion of the preoperative MRI and intraoperative CBCT and heatmap generation comparing the 3D skin surface reconstructions could be performed for 16 of 22 patients. The post hoc analysis of the impact of skin deformation on the registration accuracy revealed a considerable alteration of the skin surface model in several of the patients (Figure 7).

A post hoc analysis, in which the points acquired during the SM were matched to the intraoperative CBCT rather than the preoperative MRI, could be performed in 14 cases. The remaining eight cases had to be excluded since parts of the face were lacking on the intraoperative CBCT. An SM based on the CBCT improved the TRE by 2.8 mm on average (Figure 8). The Wilcoxon Signed-Rank test yielded a statistically significant difference in TRE for patients registered with surface matching based on CBCT compared with surface matching based on the preoperative MRI (*p* = 0.004). Furthermore, upon testing for superiority, the analysis revealed that surface matching based on CBCT was superior to surface matching on pre-op MR (*p* = 0.002).

## 4. Discussion

Today, the use of neuronavigation is standard in most intracranial surgeries. What could be considered a clinically acceptable accuracy depends on the type of surgery. While some neurosurgical procedures, such as defining the borders of a bone flap, place lower demands on accuracy, other procedures such as tumor resection in eloquent areas, frameless biopsies, or drilling of skull base structures require sub-millimeter accuracy [20]. SM is the current gold standard for patient registration [2,11,21]. It relies on the matching of facial features of the patient to preoperative 3D imaging. However, SM can be difficult to perform depending on patient positioning. When the head is turned, or the patient is prone, line-of-sight problems may occur relative to the camera of the navigation system. Facial features may also shift with gravity to create uncertainties in matching. In this study, we compared the accuracy of a standard SM method for patient registration to the accuracy of AIR based on intraoperative CBCT using the Universal AIR registration matrix. In theory, AIR has the advantage of being indifferent to patient positioning, as it relies on intraoperative imaging with the patient placed in the intended position for the surgical procedure. AIR is based on co-registering 3D positions and fiducials on the AIR device in the intraoperative image space of the patient, in the correct surgical position, to the DRF attached to the patient. Thus, the potential deformation of facial features due to patient positioning becomes completely irrelevant. We found that the accuracy of AIR was superior to that of SM.

In SM, an unexpected number of cases with a considerable TRE were noticed. Although not exclusively so, this was prominent in patients placed in prone or more laterally rotated positions. Several factors may contribute to this. As mentioned, extreme positions often lead to the obscuring of facial features from the navigation camera, precluding the use of a laser pointer (Z-touch, Brainlab, Munich, Germany) to map the face and provide positional data. Moreover, a pointer is also difficult to use in these positions. Extreme angulations of the pointer may be necessary to avoid the head obscuring the marker spheres from the view of the navigational camera. At these angles, it may not be feasible to accurately point toward the desired anatomical positions. Prone or rotated positions are also often used for posterior lesions, which are more distant from the face. As distance enhances angular errors, surface matching based on facial features is less reliable for these lesions [10,11,13,22].

In most navigated neurosurgical procedures, all accuracy hinges on the DRF. If it is dislodged, accuracy is lost. In one patient (ID 18), a 21 mm increase in TRE was found between the pre-drape and post-drape measurements, indicating a severe movement of the DRF in relation to the skull. In this case, the loss of accuracy was recognized intraoperatively, and the use of the navigation was discontinued. A retrospective analysis of this case showed that the errors of all four screws were in the same rotational plane in the Z-axis compared with the DRF, indicating a shift in the DRF position, most likely reflecting incorrect remounting after sterile draping. The overall finding that accuracy decreases after draping suggests that similar but less evident remounting errors may occur. In a clinical situation, AIR can be used after draping to avoid this error. AIR, contrary to SM, does not require access to the uncovered facial features.

Finally, using a heat map analysis, we showed that alterations in the facial features in relation to the preoperative MRI may impact the registration accuracy of SM. Facial features may shift with gravity, pull from Mayfield pins, or other factors that differ between the preoperative MRI and the intraoperative condition. TREs were generally larger when SM was used in rotated or prone positions. TREs of SM were lower when recalculated to match the CBCT performed intraoperatively, suggesting that errors were the result of differences in facial features rather than the registration procedure itself. These potential sources of error could be circumvented by using AIR, as indicated by the observed decrease in TRE.

### Strength and Limitations

A major limitation of AIR is, of course, the availability of intraoperative CT or CBCT. In this study, surgery was performed in a hybrid OR with a ceiling-mounted CBCT system. However, the registration method is independent of the imaging system and could be performed using any CT or CBCT system capable of exporting images in a standard DICOM format. Another concern would be the time dedicated to registration. Since the setup of this study required several extra steps of screw insertion and navigated point acquisition, the specific time for the registration procedure could not easily be measured. However, it is our estimate that in a dedicated environment, the registration procedure by itself could be performed in a 10-min timeframe, compared with the approximately 5 min for an SM registration [10]. Of note, the improved accuracy by AIR comes at the cost of additional radiation exposure to the patient and staff [23,24,25,26]. However, protective equipment can be used to protect OR staff, and from the perspective of the patient, a slight increase in radiation exposure may be acceptable to ensure the accuracy of a neurosurgical procedure. Moreover, the exposure from a single CBCT is lower than that of a routine CT and negligible compared with oncological adjuvant radiation.

The use of micro screws to assess accuracy was based on the assumption that their rigid fixation in the skull and their well-defined centers would help to provide the precise positional data needed for the analysis. The precision data of the point acquisition in the pairwise distance test supports this assumption. However, the use of micro screws limited the target positions to the surface of the skull, and to avoid unnecessary trauma, the screws were only placed within the intended surgical field.

## 5. Conclusions

Neuronavigation has become universally adopted in cranial neurosurgery. Part of the success rests on the use of SM to register the patient for preoperative imaging. SM is most often easy to perform and has little impact on procedural times. However, SM may be difficult to perform when the head of the patient is rotated or the patient is prone. Besides the commonly experienced line-of-sight issues that occur in these positions, our heat map analyses of skin feature shifts illustrate that inaccuracies may occur due to the stretch and pull of the soft tissues when the patient is positioned for surgery. SM is performed before sterile draping when the facial features can be visualized. Therefore, two different DRFs are used, one non-sterile and one sterile. This exchange of DRFs may introduce yet another error. The use of intraoperative 3D imaging to automatically register the patient to the navigational system avoids these difficulties and, as shown in this study, significantly improves upon the attained accuracy. Moreover, AIR is user-independent to a greater degree, resulting in consistently improved registration accuracies. An important caveat of AIR is the dependency on intraoperative imaging, which may not be readily available in all settings. Imaging based on CT or CBCT translates into increased radiation exposure, which needs to be taken into consideration.

In summary, AIR for cranial neuronavigation consistently delivered greater accuracy than SM and should be considered the new gold standard for patient registration in cranial neuronavigation. If intraoperative imaging is a limited resource, AIR should be prioritized in rotated or prone position procedures, where the benefits are the greatest.

## Figures and Tables

**Figure 1 sensors-24-07341-f001:**
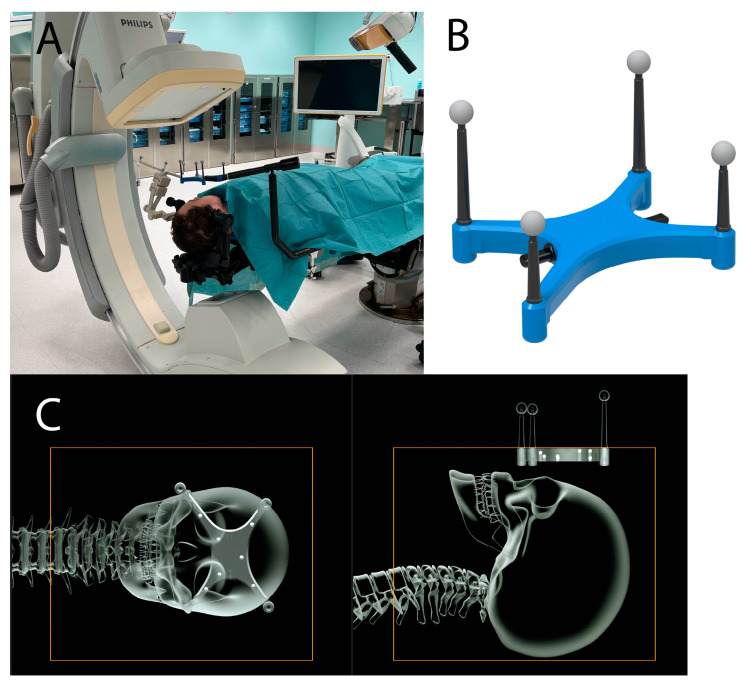
(**A**) Intraoperative setup with positioning of DRF, Universal AIR, and C-arm with patient in radiolucent head clamp. (**B**) Universal AIR matrix. (**C**) Schematic illustration of the scan volume with the Universal AIR radio-opaque markers included in the scan volume of the head.

**Figure 2 sensors-24-07341-f002:**
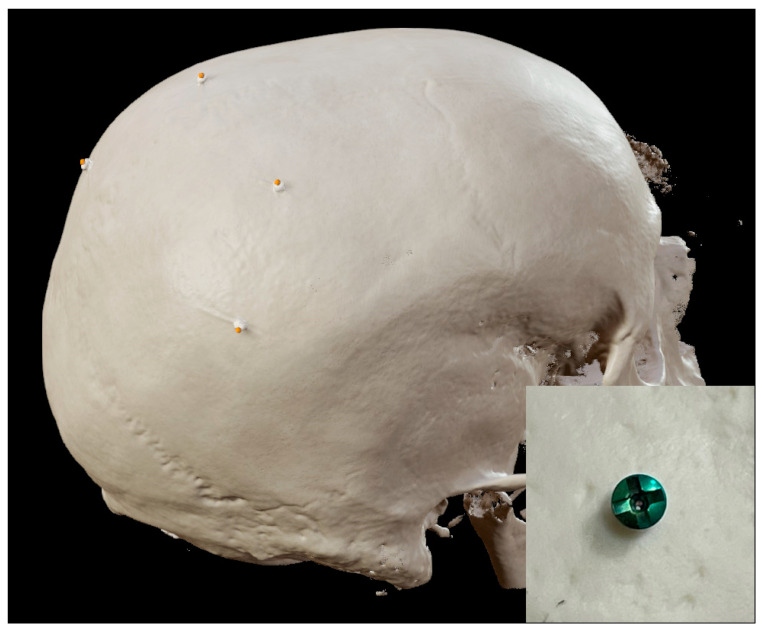
Defining the screw head reference points at the center of the slit cross hairs on 3D reconstructed CBCT imaging data. Small picture shows close-up of screw head.

**Figure 3 sensors-24-07341-f003:**
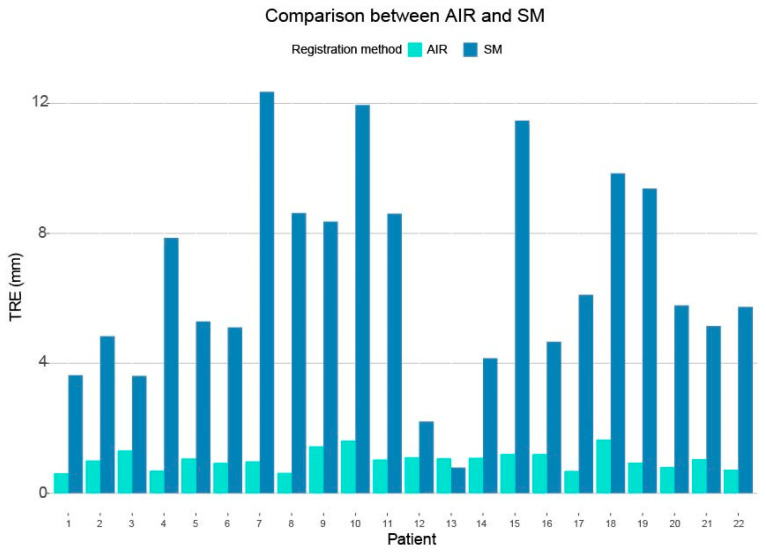
TRE calculated as mean of the difference between the acquired points using surface matching (SM) or automatic image registration (AIR) compared with the reference points for the four screws from the CBCT.

**Figure 4 sensors-24-07341-f004:**
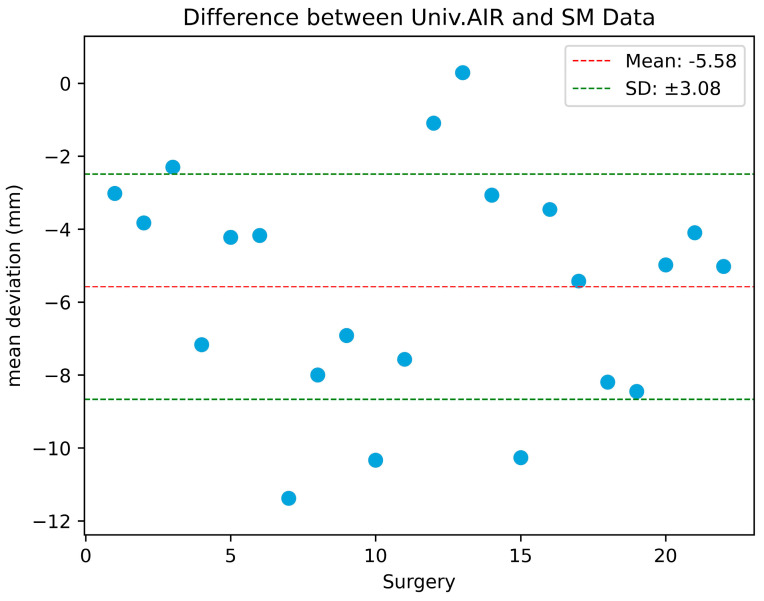
Mean difference of the calculated deviations at the four screws between AIR pre-drape and SM pre-drape registrations.

**Figure 5 sensors-24-07341-f005:**
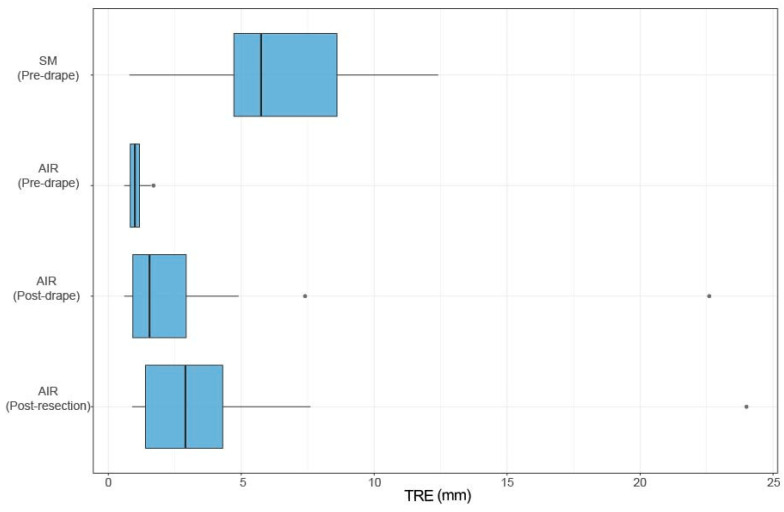
Boxplots of TRE calculated as mean of the difference between the acquired points at screw heads compared with the reference points for the four screws from the CBCT.

**Figure 6 sensors-24-07341-f006:**
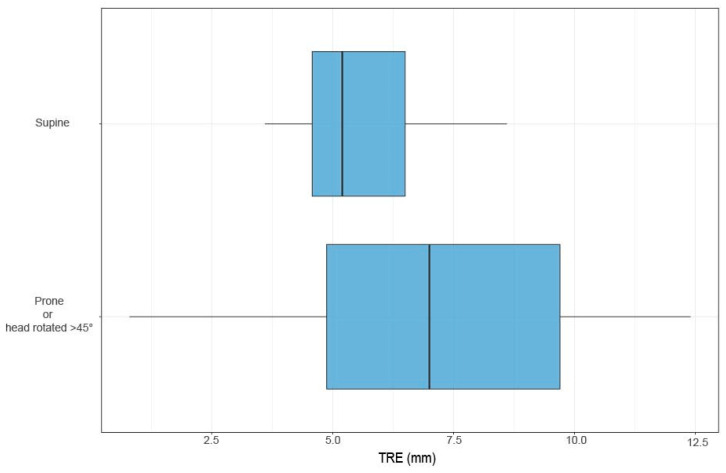
Boxplots of TRE of SM registration grouped by patient positioning.

**Figure 7 sensors-24-07341-f007:**
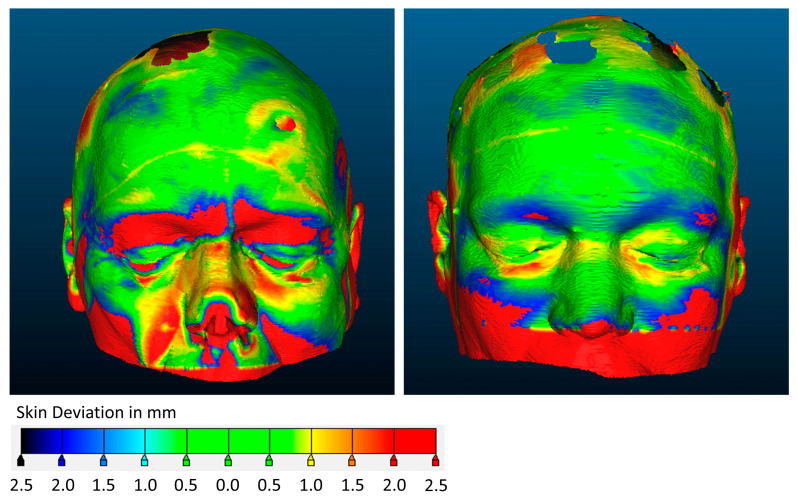
Heatmap showing skin surface deformation between the preoperative MRI used for surface match registration and the intraoperative CBCT for two patients (ID19 and 20).

**Figure 8 sensors-24-07341-f008:**
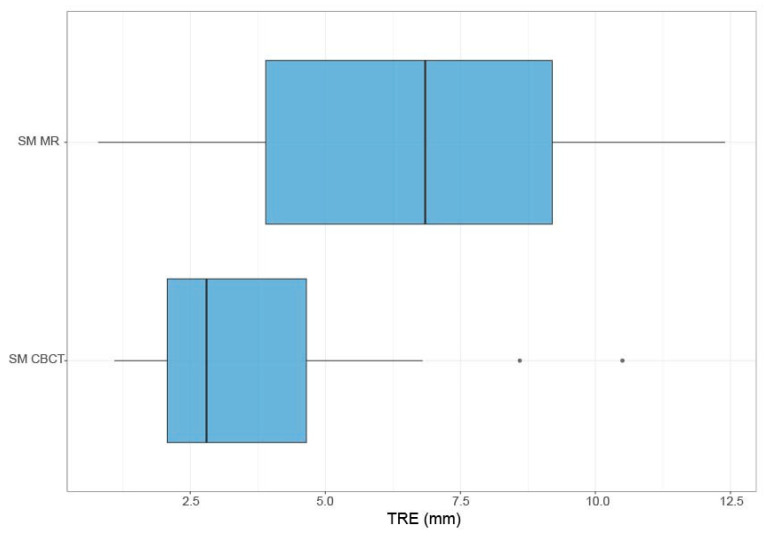
Boxplots of the TRE of the SM registration based on preoperative MRI compared with post hoc recalculated surface matching based on 3D reconstruction of the intraoperative CBCT.

## Data Availability

Data are available from the corresponding author upon reasonable request.

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
