# Peer review of "Automatic Image Registration Provides Superior Accuracy Compared with Surface Matching in Cranial Navigation"

_sensors, 2024, doi:10.3390/s24227341_

Round 1
Reviewer 1 Report
Comments and Suggestions for Authors
Bottom line, if the patient is not isn the same position in the MRI as they are planned for the surgery, there will be issues.
This fails to address the brain tissue, the real issue. "Sag" happens and what we need is accurate brain to brain locations....if possible. This will help with a lot, but there is more to do and ultimately, it is about scanning in the surgical position. Anything less will ultimately be a compromosie or require deformation, and deformation assumes things deform equally. They may not. This is novel for surgical things but is very old news for radiation oncology.
• Can a bone based system do better fusion on a non-supine patient relative to fitting to a supine image data set than a surface (skin) based system?
• In my field this is a obvious issue and is not innovative.
• What does it add to the subject area compared with other published
material?
In neurosurgery it may, but overall it does not.
• They should collect a prone image data set and see if this method is needed for prone patients – basically if just imaging in the treatment position is what is needed.
Reviewer 2 Report
Comments and Suggestions for Authors
The authors made a comparison of BrainLAB's universial AIR automatic image registration toolbox and previous surface matching method in navigation accuracy. It is beneficial to neurosurgeons and other researchers. However, there are some descriptions unclear, which needed to be improved.
1. The authors said that "The mean target registration error (TRE) for SM pre-drape was 6.6 ± 3.1 mm", which is very large, according to literature review, a neurosurgical navigation should be has a TRE of less than 2-3 mm error, or the RMSE error or FLE error to conduct neurosurgical navigation. Should the authors review some literatures and find similar results on this? The authors cited ref. 17 "Current accuracy of augmented reality neuronavigation systems: systematic review and meta-analysis", but this is a augmented reality navigation which use head-mounted display (HMD) or HUD device, which has a larger error than traditional virtual reality (VR) navigation due to these types of devices. Please cite VR navigation literatures for reference.
2. The AIR registration is not descipted in detail. I think it is a automatic image registration method and it is a rigid registration because it uses a DRF for point-based registration, so the deformation of skin induces large landmark errors, which may also occurred in AIR registration and not only in SM registration, but the authors said that AIR is more accurate, I think it is not fully believable.
3. The authors may discuss SM registration with more papers on its difference with theirs.
Round 2
Reviewer 1 Report
Comments and Suggestions for Authors
The responses from the authors and edits satisfy my concerns. I approve of this going forward.